# Hidden in Plain Tokens:
# Simply Robust, Gradient-Free Watermark for Synthetic Audio

**Georgios Milis** [1]   **Yubin Qin** [1]   **Yihan Wu** [1]   **Heng Huang** [1]

## Abstract

As policy catches up with the capabilities of generative AI, watermarking is central to content provenance efforts. Inference-time watermarks for autoregressive models are unfit for continuous modalities due to discretization inconsistencies. Existing methods overcome this by finetuning the modality tokenizers, nullifying the watermark's training-free advantage. In this work, motivated by the vocabulary redundancy of discretization, we propose an elegant solution for powerful and robust watermarking of synthetic audio. We theoretically analyze the impact of token errors on watermark detection, and effectively mitigate them using a reduced vocabulary obtained via community detection. Thorough experiments showcase that our gradient-free method can boost detectability by several orders of magnitude, while also achieving built-in robustness to audio modifications. Broadly, we discover a new state-of-the-art for token-level watermarks in multimedia, which simply arises from the nature of discrete representation learning.

## 1. Introduction

The rise of generative AI has brought attention to watermarking techniques, which imperceptibly mark AI-generated content in a way that is easily detectable algorithmically. Autoregressive models have had a central role in the development of contemporary generative AI systems. Audio models of such architecture enable the synthesis of lifelike synthetic audios, unlocking many possibilities of misuse ranging from misinformation to fraud. The development of robust imperceptible watermarks that are algorithmically detectable is a crucial requirement for addressing these concerns and restoring trust in digital media. In-generation watermarks for autoregressive models are a one-fits-all solution due to their integration at the token sampling level. Their detection requires identifying subtle statistical signals in sequences of discrete tokens retrieved from the generated content. While highly effective in discrete domains like text, the inevitable retokenization mismatch in continuous modalities seriously degrades the raw watermark signals, making detection error-prone and revealing fundamental limits of watermarking reliability.

To mitigate this, existing approaches have relied on finetuning the codec modules towards idempotence, which requires computationally expensive training and white-box access to the codec. In this work, we propose a lightweight alternative that preserves the gradient-free paradigm. We observe that retokenization errors are not random, but rather exhibit a degree of structural consistency, where tokens are mostly confused with a small set of semantic neighbors. By modeling the codec vocabulary as a graph where edges represent confusion probabilities, we can apply community detection to identify stable clusters. By applying the watermark bias at the cluster level rather than the token level, we render the watermark invariant to retokenization noise, confirming that structural alignment in the latent space translates to robustness in the signal domain. We release some audio samples and the code used for our experiments on our project page: `https://g-milis.github.io/projects/nograd-audio-wm.html`.

Our main contributions are as follows:

- We analyze the impact of retokenization on the watermark detectability from a statistical perspective, quantifying how sequence and watermark parameters contribute to the watermark's strength.

- We introduce a novel, gradient-free method that discovers the underlying vocabulary structure under retokenization and distills it before applying the watermark rule, significantly enhancing discretization stability and subsequently watermark detectability.

- We evaluate our method on models of different codec architectures and various tasks, showcasing that it is strongly detectable and robust to attacks, while minimally degrading the perceived quality.

[1]Department of Computer Science, University of Maryland, College Park, USA. Correspondence to: Georgios Milis <milis@umd.edu>, Heng Huang <heng@umd.edu>.

*Proceedings of the 43rd International Conference on Machine Learning*, Seoul, South Korea. PMLR 306, 2026. Copyright 2026 by the author(s).

## 2. Related Work

### 2.1. Audio Generative Models

Lakhotia et al. (2021) pioneered the use of discrete language modeling for audio applications, enabling models like AudioLM (Borsos et al., 2022). This discrete paradigm has also been adopted by interactive models like Kimi-Audio (Ding et al., 2025) and the streaming architecture Moshi (Défossez et al., 2024). The audio tokenizers can function as neural codecs independently of their coupled autoregressive models, for instance Moshi's Mimi codec has been adopted by LLaMA-Mimi (Sugiura et al., 2025). Broader multimodal frameworks like SEED-LLaMA (Ge et al., 2024) generalize this by interleaving audio, image, and text tokens.

### 2.2. Watermarking

**Text watermarking.** Building on the framework introduced by Aaronson (2022), Kirchenbauer et al. (2023) significantly advanced statistical watermarking and demonstrated its effectiveness through extensive experiments on large language models. The latter method partitions the vocabulary into red and green lists and biases generation toward green tokens by adding a fixed offset $\delta$ to their logits. To improve robustness, Zhao et al. (2023) proposed a unigram watermark that derives watermark keys via one-gram hashing, while Liu et al. (2023b) further enhanced robustness by using semantic features of the generated content as watermark keys. In a different direction, Liu et al. (2023a) introduced an unforgeable watermarking scheme that leverages neural networks to directly modify token distributions, rather than relying on explicit watermark keys.

**Continuous Modalities.** Tong et al. (2025) first proposed the use of a token-based watermark for images, a method fully developed later by Jovanović et al. (2025). Both approaches rely on finetuning of the discretization modules in order to reduce the retokenization errors. Wu et al. (2025a;b) identified retokenization errors in autoregressive audio models and subsequently proposed a distortion-free watermarking method based on $k$-means clustering. However, their experiments revealed lower detectability than the Kirchenbauer et al. (2023) baseline, in line with the literature from text watermarking, where strength is traded in order to achieve distortion-freeness.

**Post-hoc audio watermarking.** Audio watermarking has a long history, traditionally exploiting human insensitivity to mid-frequency bands (Lie & Chang, 2006). Recent deep-learning methods encode payloads invisibly via autoencoders in the frequency domain (Liu et al., 2024; Chen et al., 2023), with extensions for temporally localized detection (San Roman et al., 2024) and robustness via error-correcting codes (Wu et al., 2023). However, these approaches degrade audio quality, lack formal detection guarantees, are vulnerable to waveform perturbations, and require additional embedding steps. Notably, O'Reilly et al. (2025) revealed that state-of-the-art post-hoc audio watermarks can be totally erased with modern neural codecs. This highlights the need for a new watermarking paradigm, with token-based watermarks being considered as a promising direction.

## 3. Statistical Analysis

### 3.1. Preliminaries

We adopt the KGW watermarking scheme of Kirchenbauer et al. (2023). At each step, a hash of the preceding $h$ tokens partitions the vocabulary into a green set $G_i$ (with fixed ratio $\gamma$) and a red set. Watermarking is implemented by biasing the logits of green tokens with $\delta$ prior to sampling. Detection tracks the total count of green tokens, $G_{sum}$, which follows $\text{Binomial}(N, \gamma)$ under the unwatermarked null hypothesis ($H_0$). We detect the watermark using the statistic

$$z = \frac{G_{sum} - \gamma N}{\sqrt{\gamma(1-\gamma)N}}, \tag{1}$$

which for large $N$ approximately follows standard normal distribution under the null hypothesis $H_0$, showing absence of a watermark. However, the presence of a watermark would increase the number of green tokens, leading to large positive $z$ values.

### 3.2. Corruption Scenarios

Any benign or malicious modification to the generated audio waveform will affect the token sequence. In addition to that, retokenization itself introduces mismatch in continuous modalities, due to the non-idempotent nature of the encoder-decoder pair in discrete representation learning. We quantify this via the token match rate between the original sequence $x_{1:N}$ and the retokenized $y_{1:N} = E(x_{1:N})$, as

$$\text{TM}(y_{1:N}, x_{1:N}) = \frac{1}{N}\sum_{i=1}^{N}\mathbf{1}[y_i = x_i]. \tag{2}$$

While $\text{TM} \approx 1$ for text, it is significantly weaker in continuous modalities like images (Jovanović et al., 2025) or audio (O'Reilly et al., 2025), where tokenizers prioritize reconstruction quality over exact index preservation. Instead of modifying the tokenizer to enforce consistency, we address these corruptions by clustering frequently confused tokens into a reduced and robust vocabulary. We visualize the watermarked generation and retokenization of an audio sequence in Figure 1.

### 3.3. Statistical Limits of the Baseline

We model the transition from the generated watermarked sequence $x_{1:N}$ to its reconstruction $y_{1:N}$ as a noisy chan-

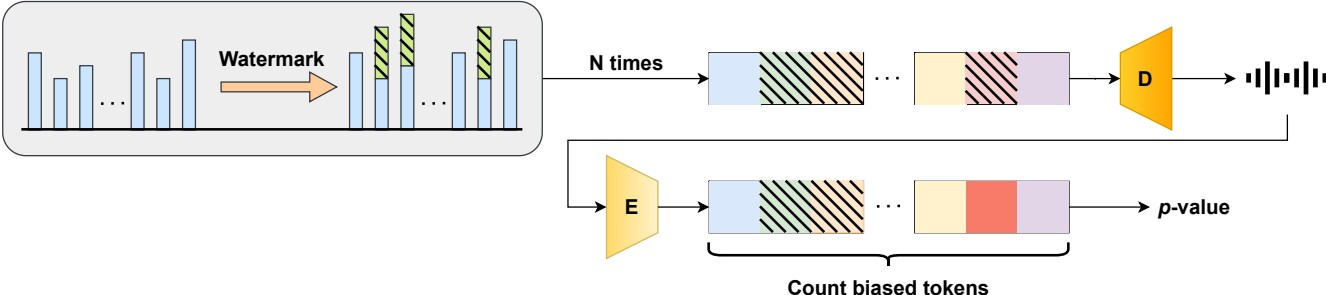

*Figure 1.* Illustration of a token-level watermarking mechanism in the audio domain. During generation, the autoregressive model computes a probability distribution over the vocabulary at each time step. A logit bias is pseudorandomly applied to a specific subset of tokens, encouraging their selection, and the resulting token sequence is synthesized into a waveform by the decoder $D$. For detection, the waveform is re-encoded by the encoder $E$ to recover the token sequence. The detector then performs a statistical hypothesis test based on the number of biased tokens, returning a $p$-value representing the probability of random generation. A sufficiently low p-value is evidence towards the alternative hypothesis, meaning that a watermark signal is present.

nel. Despite potential structural dependencies in encoder features, we assume conditional independence to derive tractable statistical limits. This assumption strictly holds for tokenizers with a non-overlapping sliding window, but we may assume that it approximately holds in general. We treat correct retokenization of tokens as independent Bernoulli events with a fixed success probability $r \in (0, 1)$, which we will later estimate empirically as the average Token Match (TM) across the dataset.

To quantify the limits of watermarking under corruption, we first derive the expected detection statistic for the standard KGW method. We assume that watermarking increases the expected proportion of green tokens from $\gamma$ to some $g > \gamma$ in the absence of corruption. Formally, under the alternative hypothesis $H_1$,

$$g = \mathbb{P}[x_i \in G_i | H_1] \tag{3}$$

is the probability that the generative model samples a token from the green partition, in other words the realized bias that the model exhibits. This value is implicitly controlled by the green list size $\gamma$ and the logit bias $\delta$.

Conditioned on a correct context, the current token $x_i$ preserves the watermark if it survives retokenization ($y_i = x_i$), which happens with probability $r$. If corrupted, it will randomly fall into the green list with probability $\gamma$. Thus, the expected probability $p_1$ of observing a green token under $H_1$ is

$$p_1 = \gamma + r^{h+1}(g - \gamma), \tag{4}$$

as detailed in Section C.1. We can calculate the expected $z$-score under $H_1$, which gives us the relationship between detectability and different variables of interest as follows

$$\mathbb{E}[z|H_1] = \sqrt{N} \frac{g - \gamma}{\sqrt{\gamma(1-\gamma)}} r^{h+1}. \tag{5}$$

This highlights an exponential decay $r^{h+1}$. Since $r < 1$ in continuous modalities, the watermark signal vanishes rapidly as the context window $h$ increases.

### 3.4. Semantic Clustering

To alleviate the high retokenization error, we introduce vocabulary clustering, and define the cluster match rate $r_{cl}$ as the probability that a retokenized token falls into the same semantic cluster as the original, formally

$$r_{cl} = \mathbb{P}[\mathcal{C}(y_i) = \mathcal{C}(x_i)], \tag{6}$$

with $\mathcal{C}$ being the clustering map. Since clusters aggregate nearest neighbors, $r_{cl} > r$. We leverage this robustness by simply replacing tokens with clusters for both the context history and biasing the current token generation. The $h$-gram context immediately becomes more robust ($r_{cl}^h$). For the current token, since the partition is defined on clusters, exact matches and cluster matches are indistinguishable. Then the expected $z$-score becomes

$$\mathbb{E}[z|H_1] = \sqrt{N} \frac{g - \gamma}{\sqrt{\gamma(1-\gamma)}} r_{cl}^{h+1}. \tag{7}$$

As $r_{cl} > r$, this effectively mitigates the exponential decay and boosts the $z$-score. We briefly extend this analysis to multiple channels in Section C.2.

### 3.5. Comparison & Trade-offs

The main contribution of this method is the improvement in the exponential base. This robustness introduces a trade-off between detectability and generative entropy. By clustering, the vocabulary size shrinks from $|V|$ to $c|V|$ for some $c \in (0, 1)$. However, the risk is key collision, especially if the key space is too small (Wu et al., 2024). Therefore, for an

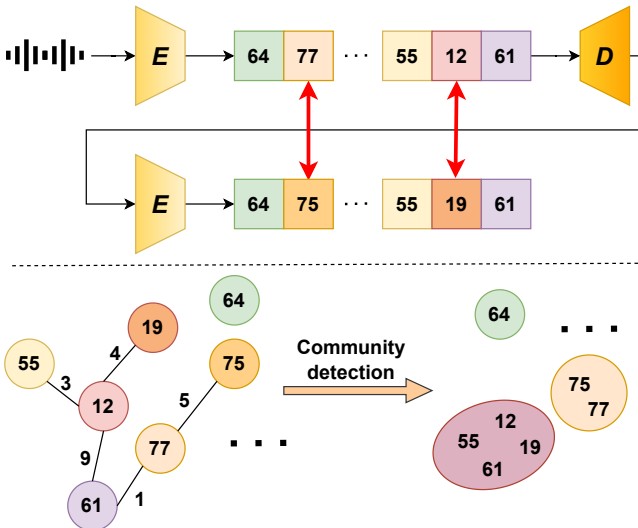

*Figure 2.* Illustration of how our method captures and explicitly mitigates the retokenization errors. First, we use the encoder and decoder modules from the codec of interest to encode, decode, and re-encode a dataset of waveforms (top). We use the confusion counts between tokens as edge weights in a graph where the vertices correspond to tokens. Then, we perform community detection on that graph, effectively reducing the vocabulary size by a many-to-one mapping from tokens to clusters (bottom). Singleton vertices represent tokens that have never been confused in the dataset. Notice that we only require black box access to the codec components.

$h$-gram context key, there should exist some threshold $K_{\min}$ such that

$$(c|V|)^h \geq K_{\min} \tag{8}$$

leads to an acceptably diverse key space. If $c$ becomes too small in order to boost $r_{cl}$, then the context length $h$ should be appropriately increased to satisfy the key space threshold. Still, the issue of handling key collisions remains, with deferral to unwatermarked sampling as a practical solution (Hu et al., 2024).

Therefore, the theoretically optimal cluster configuration must balance maximizing $r_{cl}$ for detection while maintaining a sufficiently large key space to avoid key collisions. In this work, we do not fully characterize a Pareto-optimal solution to this trade-off. Rather, we use a practical data-driven heuristic to discover strong semantic clusters, yielding $r_{cl} > r$, and show that this is sufficient for significant improvements in the watermark strength without a compromise in quality.

## 4. Methodology

In this section we present how we implement the clustering discussed in Section 3.4 via a data-driven but black-box and

gradient-free method. We observe that successful watermark detection does not necessarily require the current token $x_i$ to be retokenized correctly, it is sufficient for its retokenized counterpart $y_i$ to merely belong to the same list, either $G_i$ or $R_i$. Motivated by this, we formulate a sampling-based approach that captures the distribution of retokenization errors and appropriately distills the vocabulary via community detection. The techniques in the following subsections are applied to each RVQ channel separately, due to codebook independence.

### 4.1. Vocabulary Reduction

We obtain a dataset of paired original and retokenized token sequences, by passing an audio dataset through the codec twice. The dataset should be large enough to cover the entire token vocabulary. We then define the confusion matrix $M \in \mathbb{N}^{|V| \times |V|}$, where $|V|$ is the vocabulary size, and the entry $M_{ij}$ represents the number of times that token $i$ was confused with token $j$. We interpret $M$ as the adjacency matrix of a weighted graph $G = (V, M)$, where the node set $V$ represents the tokens.

Our goal is to find a partition $\mathcal{P} = \{c_1, c_2, \ldots, c_K\}$ that minimizes the edge weights between clusters, and maximizes the edge weights within each cluster. This is exactly the definition of modularity, thus we explore unsupervised community detection algorithms such as the Louvain (Blondel et al., 2008) and Leiden (Traag et al., 2019) methods. Both optimize for modular communities, and accept a resolution parameter $\rho$ that controls the cluster granularity.

Unlike Louvain, the Leiden method takes into account the edge directionality and provides guarantees for connected communities and faster convergence. We also empirically find that it leads to better watermark detectability, thus we encompass it into our method. Due to the multi-channel setting, we are able to enforce different resolution on different channels, leading to a strong multi-scale watermark. We visualize the vocabulary distillation process in Figure 2.

### 4.2. Cluster-Level Watermarking

In standard KGW, the token vocabulary is shuffled and split into the lists $G_i$ and $R_i$ at each time step. We simply modify this to encompass the clustering information, by shuffling and splitting the cluster vocabulary, and subsequently biasing all tokens that belong to clusters in $G_i$. Regarding the context for $h > 0$, since we require stability of the hash value, we simply replace the token indices with their cluster indices before hashing, as explained in Section 3.4. Both modifications contribute to increased watermark robustness to modifications, provided that the clustering captures well the codec's retokenization inconsistencies. This also ensures negligible computational overhead at inference time, since the only additional step (compared to standard token

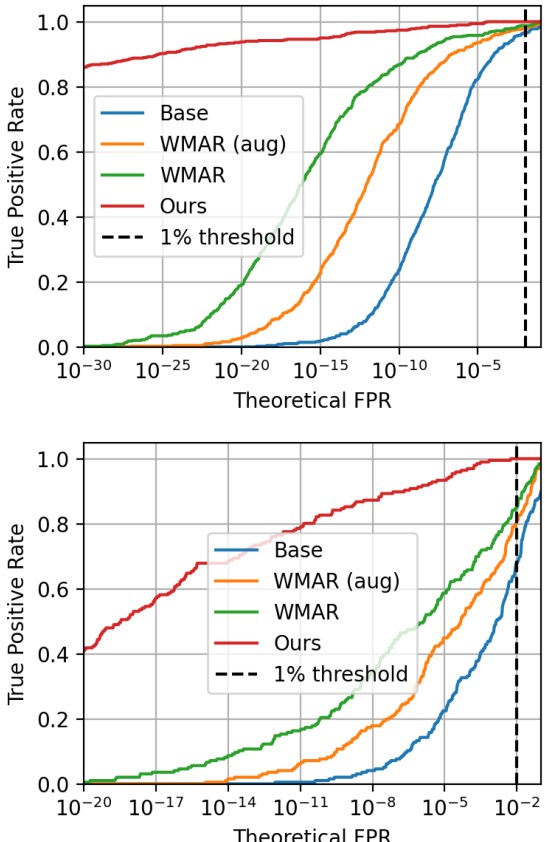

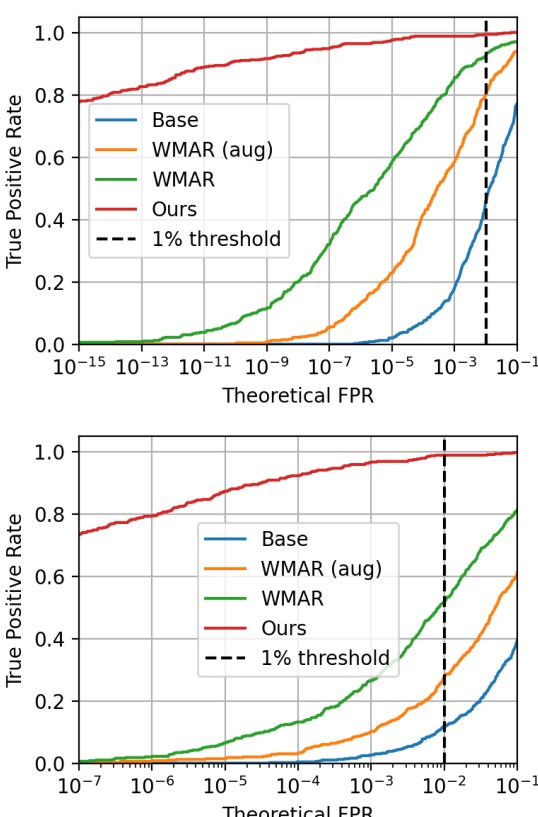

*Figure 3.* Our watermark achieves high detectability under extremely low FPR settings. Experiment on the Moshi model with $h = 0$, prompted by conversational audio inputs (top) and LibriSpeech samples (bottom).

*Figure 4.* Even with $h > 0$, our watermark still achieves unprecedented detectability even in very low FPR settings. Experiments with the Moshi model with $h = 1$ (top) and $h = 2$ (bottom), both prompted by conversational audio.

biasing) is an efficient table lookup.

The trade-offs are a reduction in the entropy of the autoregressive model distribution, as well as potential key collisions. We acknowledge that KGW already distorts the model distribution, but our experiments, as well as previous work discussed in Section 2.2 suggest that the KGW logit biasing scheme can be applied to RVQ models (with moderate bias values $\delta$), without significantly deteriorating the audio quality. We hypothesize that, unlike text, continuous modalities can afford slight distortion in their token sequences, assuming that the decoder is expressive enough to still create a meaningful waveform.

## 5. Experiments

For our main audio generation experiments, we consider two state-of-the-art audio generative models with autoregressive architecture, that span different applications and

use different audio codecs for discretization. First, we test Moshi (Défossez et al., 2024) which uses the Mimi codec and is highly capable at conversational speech. Second, we test the music generator MusicGen (Copet et al., 2023), which encompasses a 32kHz version of EnCodec (Défossez et al., 2022).

We evaluate by sampling 500 audios from each model per dataset. We prompt Moshi with the conversational prompts used in WMAR, as well as excerpts from the LibriSpeech dataset (Panayotov et al., 2015). For MusicGen, we create a custom dataset of short music descriptions by randomly shuffling descriptive words and music genres.

We compare our method against a plain KGW reweighting scheme on the raw models, referred to as "Base", as well as the two variants of the WMAR method which encompasses finetuning of the codecs towards idempotence. We present the experimental results for Moshi in the following subsections, and include experiments with MusicGen in Appendix D. In Section 5.4, we also experiment with text-to-speech

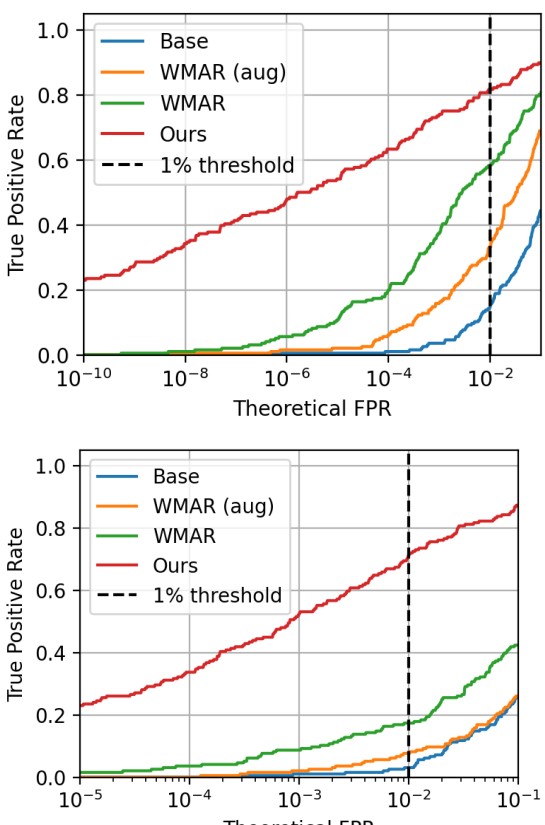

*Figure 5.* Experiments with the Moshi model with $h = 1$ (top) and $h = 2$ (bottom), both prompted by LibriSpeech samples.

*Table 1.* Audio quality scores for audios generated by Moshi prompted with conversational audio (top) and LibriSpeech (bottom).

| $h$ | Method | FAD ↓ | | MOS ↑ | |
| --- | --- | --- | --- | --- | --- |
| | | VGGish | CLAP | NISQA | DNSMOS |
| | None | 0.080 | 0.023 | $3.54 \pm 0.49$ | $4.43 \pm 0.57$ |
| 0 | Base | 0.128 | 0.020 | $3.46 \pm 0.51$ | $4.46 \pm 0.48$ |
| | WMAR (aug) | 0.267 | 0.067 | $3.43 \pm 0.50$ | $4.45 \pm 0.48$ |
| | WMAR | 0.407 | 0.032 | $3.28 \pm 0.48$ | $4.41 \pm 0.48$ |
| | Ours | 0.133 | 0.027 | $3.53 \pm 0.51$ | $4.42 \pm 0.54$ |
| 1 | Base | 0.068 | 0.014 | $3.56 \pm 0.49$ | $4.48 \pm 0.40$ |
| | WMAR (aug) | 0.218 | 0.055 | $3.54 \pm 0.48$ | $4.46 \pm 0.40$ |
| | WMAR | 0.357 | 0.024 | $3.37 \pm 0.44$ | $4.43 \pm 0.40$ |
| | Ours | 0.051 | 0.015 | $3.58 \pm 0.44$ | $4.50 \pm 0.30$ |
| 2 | Base | 0.111 | 0.021 | $3.50 \pm 0.53$ | $4.44 \pm 0.54$ |
| | WMAR (aug) | 0.189 | 0.062 | $3.47 \pm 0.52$ | $4.43 \pm 0.54$ |
| | WMAR | 0.336 | 0.030 | $3.30 \pm 0.48$ | $4.39 \pm 0.53$ |
| | Ours | 0.110 | 0.016 | $3.53 \pm 0.46$ | $4.46 \pm 0.45$ |
| $h$ | Method | FAD ↓ | | MOS ↑ | |
| | | VGGish | CLAP | NISQA | DNSMOS |
| | None | 1.921 | 0.063 | $3.15 \pm 0.77$ | $3.79 \pm 1.02$ |
| 0 | Base | 1.858 | 0.072 | $3.00 \pm 0.82$ | $3.72 \pm 1.11$ |
| | WMAR (aug) | 2.153 | 0.113 | $2.98 \pm 0.82$ | $3.69 \pm 1.11$ |
| | WMAR | 2.195 | 0.093 | $2.89 \pm 0.77$ | $3.66 \pm 1.10$ |
| | Ours | 1.670 | 0.074 | $3.07 \pm 0.77$ | $3.79 \pm 0.99$ |
| 1 | Base | 1.948 | 0.079 | $3.23 \pm 0.68$ | $4.09 \pm 0.73$ |
| | WMAR (aug) | 2.036 | 0.109 | $3.19 \pm 0.68$ | $4.06 \pm 0.74$ |
| | WMAR | 2.018 | 0.092 | $3.12 \pm 0.66$ | $4.03 \pm 0.73$ |
| | Ours | 1.921 | 0.074 | $3.22 \pm 0.62$ | $4.03 \pm 0.77$ |
| 2 | Base | 1.966 | 0.073 | $3.11 \pm 0.76$ | $3.81 \pm 1.02$ |
| | WMAR (aug) | 1.985 | 0.096 | $3.10 \pm 0.77$ | $3.78 \pm 1.02$ |
| | WMAR | 2.089 | 0.089 | $3.01 \pm 0.72$ | $3.75 \pm 0.99$ |
| | Ours | 1.823 | 0.059 | $3.15 \pm 0.75$ | $3.80 \pm 1.00$ |

as a sequence-to-sequence audio task.

## 5.1. Detectability

We evaluate detectability by showing the true positive rate (TPR) by thresholding the $p$-values at a desired false positive rate (FPR). This allows us to visualize the sensitivity of the watermark in very low FPR scenarios. We present some results for $h = 0$ in Figure 3. For higher-order $h$-grams, the results are presented in Figure 4 and Figure 5. We attribute the generally worse performance on LibriSpeech due to the Moshi model's training distribution and scope as a conversational model, while LibriSpeech prompts are audiobook excerpts.

All figures showcase that our watermark is several orders of magnitude stronger than the baselines, even finetuned methods like WMAR. The distilled vocabulary is extremely effective at alleviating the errors, which contribute to discounted green tokens not only in the watermark stream, but also in the context used to determine the pseudo-randomness.

## 5.2. Audio Quality

The superior detectability of our watermark, especially given that it relies on KGW, which is not unbiased in terms of sequence modeling, raises the question of whether it compromises the audio quality of the generated samples.

Given the in-generation nature of our watermark and the baselines, we rely on non-intrusive (reference-free) methods for quality evaluation. We use the statistical Fréchet Audio Distance (FAD) (Kilgour et al., 2019), computing the FAD score between sets of watermarked audios and their unwatermarked counterparts generated from identical prompts. We employ two different feature extractors under the hood, namely the convolutional VGGish (Hershey et al., 2017) and the transformer-based CLAP (Elizalde et al., 2023). We also report the predicted mean opinion score (MOS) from the NISQA (Mittag et al., 2021) and DNSMOSPro (Cumlin et al., 2024) models. We present the full results in Table 1, demonstrating that our method does not severely impact the audio quality compared to the baselines.

Our method consistently outperforms the baselines in every

*Table 2.* Mean $-\log p$-values and Loss (relative to Identity) for Moshi using conversational audio (top) and LibriSpeech (bottom).

| Group | Transformation | Base $-\log(p)$ | Base Loss | WMAR $-\log(p)$ | WMAR Loss | WMAR (aug) $-\log(p)$ | WMAR (aug) Loss | Ours $-\log(p)$ | Ours Loss |
|---|---|---|---|---|---|---|---|---|---|
| Baseline | Identity | 8.51 | 0.00 | 17.44 | 0.00 | 13.72 | 0.00 | 42.47 | 0.00 |
| Signal Proc. | Smooth | 1.99 | 6.52 | 1.61 | 15.84 | 3.73 | 9.99 | 32.68 | 9.80 |
| | Lowpass | 5.82 | 2.69 | 9.23 | 8.21 | 10.52 | 3.20 | 41.51 | 0.96 |
| | Highpass | 1.11 | 7.40 | 1.50 | 15.95 | 1.94 | 11.79 | 25.59 | 16.89 |
| | Noise | 2.23 | 6.28 | 0.61 | 16.83 | 8.01 | 5.71 | 20.59 | 21.89 |
| Compression | MP3 | 7.47 | 1.04 | 15.31 | 2.14 | 12.66 | 1.06 | 41.26 | 1.22 |
| | DAC | 6.62 | 1.89 | 8.12 | 9.32 | 10.51 | 3.22 | 40.13 | 2.35 |
| | EnCodec | 2.59 | 5.92 | 2.82 | 14.62 | 2.78 | 10.95 | 32.64 | 9.84 |
| | SpeechTok | 4.48 | 4.03 | 4.29 | 13.15 | 4.28 | 9.44 | 35.92 | 6.55 |
| | FaCodec | 4.73 | 3.77 | 5.36 | 12.08 | 4.75 | 8.97 | 38.47 | 4.00 |
| Temporal | Crop | 1.51 | 7.00 | 1.27 | 16.18 | 1.51 | 12.22 | 16.48 | 26.00 |
| | Shift | 1.86 | 6.65 | 1.81 | 15.63 | 2.36 | 11.37 | 27.67 | 14.80 |
| | Speedup | 1.52 | 6.99 | 1.20 | 16.25 | 1.35 | 12.37 | 26.49 | 15.99 |
| Baseline | Identity | 3.61 | 0.00 | 7.95 | 0.00 | 5.98 | 0.00 | 22.77 | 0.00 |
| Signal Proc. | Smooth | 1.50 | 2.11 | 1.22 | 6.72 | 2.02 | 3.96 | 16.96 | 5.82 |
| | Lowpass | 2.99 | 0.61 | 4.60 | 3.35 | 4.86 | 1.11 | 21.05 | 1.72 |
| | Highpass | 0.97 | 2.63 | 1.16 | 6.79 | 1.69 | 4.29 | 14.29 | 8.49 |
| | Noise | 1.49 | 2.12 | 0.47 | 7.48 | 3.74 | 2.23 | 9.19 | 13.58 |
| Compression | MP3 | 3.30 | 0.30 | 7.27 | 0.68 | 5.67 | 0.31 | 22.29 | 0.49 |
| | DAC | 3.10 | 0.50 | 4.19 | 3.76 | 4.90 | 1.08 | 22.10 | 0.68 |
| | EnCodec | 1.65 | 1.96 | 1.78 | 6.16 | 1.79 | 4.18 | 17.56 | 5.22 |
| | SpeechTok | 2.43 | 1.18 | 2.46 | 5.49 | 2.27 | 3.70 | 19.50 | 3.28 |
| Temporal | Crop | 1.24 | 2.37 | 1.24 | 6.71 | 1.39 | 4.58 | 9.70 | 13.08 |
| | Shift | 1.36 | 2.24 | 1.51 | 6.44 | 1.79 | 4.19 | 16.09 | 6.68 |
| | Speedup | 1.06 | 2.55 | 1.07 | 6.88 | 1.23 | 4.75 | 15.54 | 7.23 |

*Table 3.* Audio quality in the text-to-speech setting, with audios generated by the CosyVoice3 and Spark-TTS models.

| Model | Method | FAD ↓ VGGish | FAD ↓ CLAP | MOS ↑ NISQA | MOS ↑ DNSMOS | ASR ↓ WER | ASR ↓ CER |
|---|---|---|---|---|---|---|---|
| CosyVoice3 | None | 0.0964 | 0.0235 | 3.86 | 2.73 | 0.0323 | 0.0178 |
| | Base | 0.1954 | 0.0267 | 3.75 | 2.70 | 0.0586 | 0.0366 |
| | Ours | 0.1942 | 0.0294 | 3.81 | 2.74 | 0.0519 | 0.0308 |
| Spark-TTS | None | 0.2057 | 0.0401 | 3.37 | 2.94 | 0.0097 | 0.0018 |
| | Base | 0.2221 | 0.0484 | 3.30 | 2.97 | 0.0098 | 0.0012 |
| | Ours | 0.3506 | 0.0472 | 3.46 | 2.96 | 0.0099 | 0.0024 |

attack scenario, although all token-based methods are reasonably robust to modifications that do not cause temporal misalignment. Such modifications, like cropping or speed changes, can inherently break token-level watermarks. We address this limitation in Appendix B.

### 5.3. Robustness

A watermark's robustness to modifications is crucial for practical applications, since even benign modifications can be detrimental to the watermarking signal. Codecs are ubiquitously used for efficiency, and everyday users use editing

*Table 4.* Watermark detectability metrics for the CosyVoice3 and Spark-TTS models.

| Model | Method | $p \downarrow$ | $-\log(p) \uparrow$ |
|---|---|---|---|
| | None | 0.1885 | 0.863 |
| Cosyvoice3 | Base | 0.03394 | 1.564 |
| | Ours | $4.89 \cdot 10^{-14}$ | 13.927 |
| | None | 0.4953 | 0.400 |
| Spark-TTS | Base | $2.061 \cdot 10^{-9}$ | 9.237 |
| | Ours | $5.466 \cdot 10^{-18}$ | 17.806 |

tools built into social media platforms to edit their content for viewership purposes. Furthermore, users may maliciously attempt to post-process AI-generated audios to erase potential watermarks, even if they have no knowledge of the watermarking mechanism.

To evaluate the robustness of our method under realistic conditions, we apply a diverse suite of transformations, including signal processing modifications (smoothing, low-pass filtering, highpass filtering, noise addition), diverse audio codecs (MP3, DAC (Kumar et al., 2023), EnCodec (Défossez et al., 2022), SpeechTokenizer (Zhang et al., 2023), and FaCodec (Ju et al., 2024)), and temporal modifications (cropping, shift, speedup). The detailed settings are presented in Appendix E.

The results are summarized in Table 2, while additional results are included in the Appendix D. We report the average $-\log p$-value for each setting, as well as the reduction (Loss) relative to the identity, to highlight which attacks have a stronger impact on the token-level watermarks. We emphasize that the 1% FPR threshold corresponds to $-\log p = 2$.

### 5.4. Extension to Flow-Enhanced Text-to-Speech

We examine our method's generalization to a sequence-to-sequence task rather than pure generative modeling. We perform a case study on mainstream text-to-speech (TTS) models. Specifically, we examine whether our method can generalize to state-of-the-art models with autoregressive core and subsequent flow-matching for acoustic refining. In this paradigm, flow matching can be considered part of the decoder shown in Figure 2.

Our method relies on the effective clustering of retokenization errors, which are also available in such architectures. We experiment with CosyVoice3 (Du et al., 2025) and Spark-TTS (Wang et al., 2025), using the conversational prompts as text input. We present the watermark detectability in Table 4 as median $p$-values and average $-\log p$, demonstrating orders of magnitude better detectability than the KGW method. We present the corresponding audio quality in Table 3, where we also include automatic speech recognition (ASR) errors using the Whisper model (Radford et al., 2022).

These results suggest that our method is highly versatile, generalizing to the flow-matching enhancement paradigm, without severely compromising quality.

## 6. Conclusion

In this work, we addressed the fundamental instability of inference-time watermarking for autoregressive audio generation. Rather than resorting to invasive and costly codec finetuning of the discretization module towards idempotence, we propose an elegant, lightweight, and gradient-free alternative that exploits the intrinsic redundancy of the discrete vocabulary. By modeling the retokenization errors as stochastic transitions in a graph topology, we derive robust token communities that significantly alleviate the discretization mismatch. Our evaluations on state-of-the-art models spanning different audio generative tasks, demonstrate that this simple structural intervention yields unprecedented detectability, without degrading generation quality. Ultimately, our findings suggest that the perceived fragility of token-level audio watermarking is solvable by merely distilling the models' learned discrete representation space.

## Acknowledgements

This work was partially supported by NSF IIS 2347592, 2348169, DBI 2405416, CCF 2348306, CNS 2347617, RISE 2536663.

## Impact Statement

Our work provides a valuable contribution to the field of watermarking for AI-generated content, which is essential for provenance efforts and trust in digital media. AI governance requires transparency and accountability, and explicitly marking synthetic content in a statistically robust way is extremely useful. For instance, given adoption by large model providers, the watermark detection mechanism can be deployed in social media in order to automatically flag synthetic content. Moreover, a very low $p$-value may serve as sufficient grounds to dismiss fabricated evidence in court. Potential negative societal impact would be concerns regarding surveillance, loss of anonymity, and non-consensual tracking of a user's digital activity. Furthermore, deployment of watermarking without clear disclosure policies could lead to erosion of trust in generative AI tools. Altogether, the impacts highlight the importance of ethical deployment, transparency, and user literacy regarding provenance methods.

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

## A. Extensibility

Our vocabulary distillation method could in principle be applied to other multimodal autoregressive models, such as Emu3 (Wang et al., 2024). Since we consider the RVQ architecture of modern audio codecs, where each channel typically has a much smaller vocabulary size than image discretizers, we hypothesize that an extension to the image domain could require different, perhaps much more diverse sampling of the codec, or tailored hyperparameter settings in order to be effective. Indeed, Jovanović et al. (2025) already acknowledge that some image codebooks suffer from low utilization, requiring stratification to balance the red and green lists. We thus choose to focus on audio and provide a comprehensive audio-specific evaluation, leaving applications with different modality-specific technicalities for future work.

## B. Limitations

Our watermark's main limitation is its brittleness to temporal modifications, which are common in social media settings, especially in short form content where cropping, speedup, or slowdown are typically used for user engagement. We argue that this limitation is inherent to any token-based in-generation watermark scheme due to a misaligned waveform being separated into non-overlapping frames, leading to different tokenization. However, it does not negate the superiority over post-hoc watermarks in sophisticated codecs, and is addressable via post-processing.

For instance, cropping can be easily bypassed by zero-padding the start of the waveform by uniformly spaced crop offsets between zero and the tokenizer's window size, performing detection on all padded versions. If the waveform is indeed watermarked but cropped, padding by the right amount will align it back to the original tokenization. A similar linear search strategy can be applied with different speed modifications that resynchronize the modified audio by matching it back to the original speed. A lightweight speed estimator trained on the adopted codec would also be a viable solution for industry practitioners. Furthermore, watermark synchronization methods can be used for explicitly encoding alignment information, such as a post-hoc watermark modulated by a square wave (Jovanović et al., 2025). Policymakers and industry practitioners may decide to use an ensemble of different watermarking approaches to ensure robustness to any type of in-the-wild modifications.

Finally, while our method is gradient-free, capture and clustering of the retokenization errors is required. Thus, adaptation to a new model still requires some light computational work, as well as black-box access to the model's tokenizer.

## C. Extended Theoretical Analysis

### C.1. Green Tokens After Corruption

Under the null hypothesis, $H_0$, the expected percentage of green tokens in a generated sequence would be $\gamma$. Then, the expected ratio of green tokens after retokenization would be

$$\gamma r + \gamma(1 - r)\gamma + (1 - \gamma)(1 - r)\gamma = \gamma. \tag{9}$$

Under $H_1$, assuming correct context $h$, and with an expected ratio of green tokens $g > \gamma$, we would have

$$gr + g(1 - r)\gamma + (1 - g)(1 - r)\gamma = gr + (1 - r)\gamma = \gamma + r(g - \gamma). \tag{10}$$

If the context is correct, which happens with probability $r^h$, the expectation of green tokens is $r^h(\gamma + r(g - \gamma))$. If it's corrupted, with probability $(1 - r^h)$, the partition becomes random with green probability of $\gamma$. Therefore, the final expected ratio is

$$r^h(\gamma + r(g - \gamma)) + (1 - r^h)\gamma = \gamma + r^{h+1}(g - \gamma), \tag{11}$$

which is exactly Equation (4).

### C.2. Extension to Multi-Channel

To formalize the benefit of our proposed clustering approach in this a multi-channel setting like RVQ, we derive the expected z-score for the combined multi-channel statistic.

For a single channel $c$, the expected shift in the green count is determined by the robustness of the matching mechanism. The signal is preserved if the retokenized vector falls into the correct semantic cluster. The total statistical deviation is the

*Table 5.* Theoretical and empirical $z$-scores with correlation coefficients for different $h$.

| $\gamma$ | $h = 0$ | | $h = 1$ | | $h = 2$ | |
|---|---|---|---|---|---|---|
| | Theoretical | Empirical | Theoretical | Empirical | Theoretical | Empirical |
| 0.1 | 6.1123 | 6.7318 | 4.9797 | 9.2911 | 4.0570 | 7.6467 |
| 0.2 | 7.0953 | 7.0225 | 5.7805 | 7.5169 | 4.7094 | 6.8360 |
| 0.3 | 6.8406 | 6.8239 | 5.5730 | 7.5557 | 4.5404 | 6.0868 |
| 0.4 | 6.3152 | 5.6013 | 5.1450 | 6.3814 | 4.1916 | 5.7638 |
| 0.5 | 5.9394 | 5.3666 | 4.8389 | 5.6507 | 3.9422 | 4.6116 |
| 0.6 | 5.0018 | 4.5129 | 4.0749 | 4.9890 | 3.3199 | 4.2382 |
| 0.7 | 4.3240 | 4.1575 | 3.5228 | 3.9197 | 2.8700 | 2.9864 |
| 0.8 | 3.4383 | 3.5239 | 2.8012 | 3.0123 | 2.2821 | 2.3622 |
| Correlation | 0.9465 | | 0.8535 | | 0.8872 | |

sum across all channels

$$\sum_{c=1}^{C} \mathbb{E}[G_{sum,c} - \mu_0] = \sum_{c=1}^{C} N(g_c - \gamma)(r_{cl,c})^{h+1}. \tag{12}$$

Due to independence across channels, the variances sum linearly as

$$\sigma_{\text{total}}^2 = \sum_{c=1}^{C} \sigma_0^2 = CN\gamma(1 - \gamma). \tag{13}$$

Thus, much like in the single-channel case, the expected statistic $\mathbb{E}[z_{\text{total}}]$ is the total signal standardized by the total variance

$$\mathbb{E}[z_{\text{total}}] = \frac{\sum_{c=1}^{C} N(g_c - \gamma)(r_{cl,c})^{h+1}}{\sqrt{CN\gamma(1 - \gamma)}}, \tag{14}$$

which is the statistic we empirically measure through the actual $G_{sum,c}$ of every channel, and use it to report the watermark $p$-value.

### C.3. Validation of Analysis

In order to strengthen the validity of our analysis, we empirically verify the expected $z$-score for single-channel watermarking (derived in Section 3.3). We use MusicGen, since we were able to embed robust watermarks on only one of its RVQ channels (unlike Moshi, where watermarking on only one of the 8 channels was not strong). First, we estimate $r$ and $g$ from an independent set of samples using Equations 2 and 3. Then we proceed to estimate Equation 5 for a fixed generation length $N$, comparing it with the actual empirical $z$-score from a set of samples (independent from the previous ones) in Table 5. The high correlation indicates that our modeling, while simplified, captures very well the interplay of token-level watermarking with an unstable tokenizer.

Thus, boosting detectability via artificially increasing $r$ via clustering is theoretically sound. Unfortunately, the clustering effect is not as easily predictable, and we are unable to verify Equation 7 with equally high correlation for all values of $\gamma$. We attribute this to $g$ (the aggregate effect of watermarking) implicitly relying on the clustering map used, and thus being intractable to model. However, our derivation is still valid as a tractable simplified model indicating how watermark parameters are expected to impact the detectability.

### C.4. Clustering Effectiveness

We finally examine how the clustering-based vocabulary reduction actually tackles the retokenization errors, leading to a strong watermarking signal. The variable of interest is $r_{cl}$, defined in Section 3.4 as the probability of a token maintaining its cluster identity after retokenization, as compared to the baseline $r$ which measures the token match. In other words, $r_{cl}$ is the cluster-level match that bypasses the brittleness of the token-level resolution. In Figure 6, we plot $r_{cl}$ across hyperparameter sweeps of the resolution $\rho$ and the noise threshold for the Leiden community detection algorithm in the first Mimi channel.

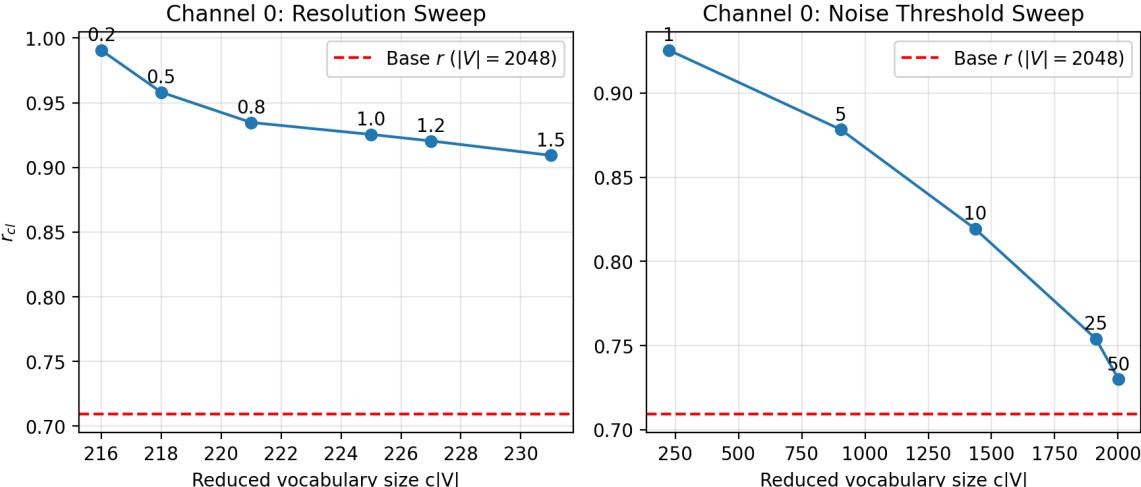

*Figure 6.* Clustering effectiveness of the Leiden community detection method for different hyperparameter sweeps. As expected, the cluster preservation is much stronger than the baseline token preservation $r$. We observe that the effectiveness increases in small resolutions (where larger clusters are encouraged, thus fewer chances of inter-cluster error), and in small noise thresholds (meaning that even rare token substitutions are important). However, both of these directions also decrease the vocabulary size measured on the horizontal axis.

Despite the effectiveness being optimal in small resolutions and small noise thresholds, the decrease in the vocabulary size can lead to distorted output and key collisions. We thus perform a per-channel selection of hyperparameters by activating the watermark only on the tokens of one channel, performing a grid search, and selecting the optimal configurations.

*Table 6.* Mean $-\log p$-values and Loss (relative to Identity) for MusicGen with $h = 0$.

| Group | Transformation | Base $-\log(p)$ | Base Loss | WMAR $-\log(p)$ | WMAR Loss | Ours $-\log(p)$ | Ours Loss |
|---|---|---|---|---|---|---|---|
| Baseline | Identity | 9.05 | 0.00 | 9.06 | 0.00 | 28.46 | 0.00 |
| Signal Proc. | Smooth | 1.01 | 8.04 | 0.85 | 8.21 | 14.39 | 14.08 |
| | Lowpass | 6.31 | 2.74 | 5.47 | 3.59 | 25.55 | 2.91 |
| | Highpass | 1.27 | 7.78 | 1.36 | 7.70 | 0.72 | 27.74 |
| | Noise | 6.88 | 2.16 | 5.53 | 3.53 | 24.92 | 3.54 |
| Compression | MP3 | 8.98 | 0.07 | 8.94 | 0.12 | 27.86 | 0.60 |
| | DAC | 1.67 | 7.37 | 1.78 | 7.28 | 19.27 | 9.19 |
| | SpeechTok | 0.98 | 8.07 | 0.96 | 8.09 | 13.75 | 14.72 |
| | FaCodec | 0.62 | 8.43 | 0.62 | 8.44 | 14.92 | 13.54 |
| Temporal | Crop | 0.93 | 8.11 | 1.10 | 7.96 | 11.73 | 16.73 |
| | Shift | 0.69 | 8.35 | 0.88 | 8.17 | 16.41 | 12.05 |
| | Speedup | 0.51 | 8.54 | 0.72 | 8.34 | 9.66 | 18.81 |

## D. Supplementary Experiments

We experiment with the MusicGen model in a caption-to-music task, and show the results in Figure 7. Table 7 shows the quality metrics obtained in the music generation task. We only evaluate with the FAD metrics, since MOS is tailored for speech. Finally, robustness experiments for the music task are shown in Table 6.

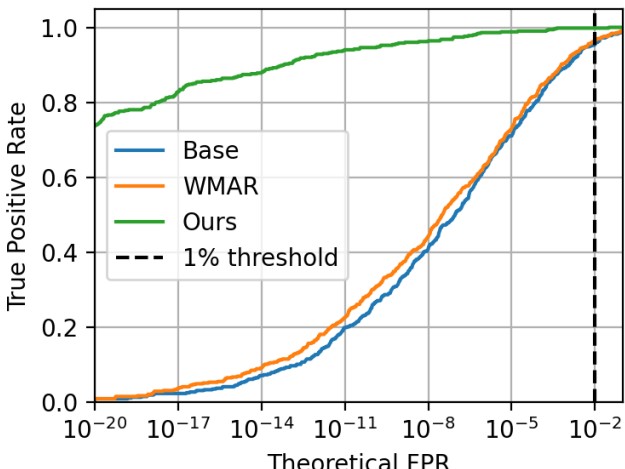

*Figure 7.* Experiments with the MusicGen model with $h = 0$, prompted by captions describing music. Our proposed method is still superior to the baselines despite the different architecture and task.

*Table 7.* Audio quality scores for audios generated by MusicGen with music prompts.

| $h$-gram | Method | FAD ↓ | |
| | | VGGish | CLAP |
|---|---|---|---|
| | None | 0.247 | 0.039 |
| $h = 0$ | Base | 0.330 | 0.043 |
| | WMAR | 1.193 | 0.132 |
| | Ours | 1.256 | 0.082 |

# E. Experimental Details

We first create the vocabulary reductions for our proposed method by performing community detection on the retokenization errors of audios. For the Mimi codec and the TTS models in the flow-matching case study, we use 2.7k audios from the full development set of LibriSpeech, while for EnCodec we use 2.1k music segments from the MusicCaps dataset (Agostinelli et al., 2023). To account for potential noise and search for more robust communities, we perform a grid search of clustering hyperparameters for each channel.

Namely, we perform experiments on a validation set with single channel watermarks (keeping the other channels unwatermarked) with different $\rho$ and a noise threshold $m$ that filters small entries in the adjacency matrix $M$, and eventually keep the pair that results in the highest detectability without deviating too much from the pair $(\rho, m) = (1, 1)$. We empirically make sure that each configuration does not result in very large monolithic clusters or a low cluster vocabulary. We present these empirical selections in Table 8 for reproducibility, and present the full ablation results in Table 9. Thus, we select the optimal configuration per channel, and finally apply watermarking to all channels, each with its own optimal clustering.

We use the finetuned Moshi checkpoints released by WMAR, but for MusicGen we finetune the 32kHz version of EnCodec. To finetune MusicGen's codec, we use a balanced mixture of music and speech data from Free Music Archive (Defferrard et al., 2017) and LibriTTS (Zen et al., 2019), respectively. We use the `medium` sized checkpoint for generation, and finetune in the non-augmented setting.

As watermarking parameters, we follow Jovanović et al. (2025) and use $\gamma = 0.25$ everywhere and $\delta = 2$ for Moshi and Spark-TTS. We empirically attenuated the watermark strength to achieve a better detectability-quality trade-off, namely we used $\delta = 1$ for MusicGen, and $\delta = 0.5$ for CosyVoice3. Regarding $h > 0$, we choose to use the corresponding $h$-gram from the first channel as a hash key at each time step, since the lower channels have better robustness to retokenization. We set the generation length to 200, which leads to roughly 10 sec of audio for Moshi, and 4 sec of audio for MusicGen. As in

*Table 8.* Selected hyperparameters $(\rho, m)$ for each model, channel-specific where applicable.

| Model (codec) | Channel | Selected pair |
|---|---|---|
| Moshi (Mimi) | 0 | (0.8, 1) |
| | 1 | (0.8, 10) |
| | 2 | (0.8, 10) |
| | 3 | (0.8, 1) |
| MusicGen (EnCodec) | 0 | (0.8, 1) |
| | 1 | (0.8, 1) |
| | 2 | (1.2, 1) |
| | 3 | (1.2, 1) |
| CosyVoice3 | 0 | (0.8, 1) |
| Spark-TTS (BiCodec) | 0 | (1.0, 10) |

Jovanović et al. (2025), we apply the watermark on the first 4 audio streams of Moshi, leaving the semantic (text) stream untouched. Similarly, we apply the watermark on all 4 streams of MusicGen.

For the robustness experiments, we use the following diverse attack suite:

- **Speed perturbation:** Resamples the audio to increase playback speed by a factor of 1.1.

- **Noise injection:** Adds white Gaussian noise with a standard deviation of $\sigma = 0.01$.

- **Lowpass filter:** Attenuates frequencies above a cutoff of 3 kHz.

- **Highpass filter:** Attenuates frequencies below a cutoff of 1 kHz.

- **Signal smoothing:** Applies a smoothing filter to reduce high-frequency variations.

- **MP3 compression:** Applies standard lossy compression at a bitrate of 64 kbps.

- **Time shift:** Shifts the waveform start time by fractional frame offsets ($1/8$ and $1/2$ of a frame) to test synchronization.

- **Temporal crop:** Truncates the audio to retain only $50\%$ of the original duration.

- **Neural compression:** Transcodes the output using external neural codecs (DAC, Encodec, SpeechTokenizer, and FaCodec).

*Table 9.* Clustering metrics and $p$-values evaluations across different channels.

| Channel | $(\rho, m)$ | Max cluster size | No. of clusters | $-\log(p)$ |
|---|---|---|---|---|
| $c = 0$ | (0.8, 1) | 883 | 151 | 39.049 |
| | (1.0, 5) | 182 | 659 | 29.025 |
| | (0.8, 5) | 247 | 655 | 26.946 |
| | (1.0, 10) | 121 | 1071 | 14.836 |
| | (1.2, 10) | 89 | 1072 | 14.517 |
| | (1.2, 5) | 153 | 664 | 12.427 |
| | (0.5, 10) | 165 | 1067 | 10.731 |
| | (0.5, 5) | 349 | 649 | 9.601 |
| | (1.2, 1) | 325 | 164 | 9.412 |
| | Base | 1 | 2048 | 9.044 |
| $c = 1$ | (0.5, 1) | 1795 | 254 | 78.916 |
| | (0.8, 1) | 1611 | 256 | 68.069 |
| | (1.2, 5) | 206 | 722 | 17.506 |
| | (0.8, 10) | 128 | 1371 | 16.086 |
| | (0.5, 5) | 646 | 727 | 13.475 |
| | (1.2, 10) | 103 | 1374 | 13.322 |
| | (1.0, 5) | 201 | 718 | 10.059 |
| | Base | 1 | 2048 | 9.876 |
| $c = 2$ | (1.2, 5) | 330 | 880 | 29.952 |
| | (0.8, 5) | 394 | 899 | 25.835 |
| | (0.8, 10) | 164 | 1552 | 22.198 |
| | (1.2, 1) | 421 | 375 | 17.655 |
| | (1.2, 10) | 85 | 1536 | 17.185 |
| | (1.0, 5) | 391 | 889 | 14.934 |
| | (1.0, 10) | 119 | 1540 | 13.391 |
| | Base | 1 | 2048 | 12.099 |
| | (0.5, 10) | 282 | 1548 | 8.505 |
| $c = 3$ | (1.0, 5) | 427 | 947 | 31.253 |
| | (1.2, 5) | 306 | 950 | 20.810 |
| | (0.8, 10) | 116 | 1635 | 16.888 |
| | (1.2, 10) | 56 | 1628 | 15.843 |
| | Base | 1 | 2048 | 15.230 |
| | (0.5, 10) | 308 | 1640 | 10.961 |
| | (1.2, 1) | 691 | 341 | 10.609 |
| | (0.8, 5) | 469 | 946 | 9.493 |
| | (1.0, 10) | 122 | 1626 | 7.657 |

