# OpenReview forum: "Hidden in Plain Tokens: Simply Robust, Gradient-Free Watermark for Synthetic Audio"
_ICML.cc/2026/Conference — ICML 2026 regular_

### Official Review · Reviewer_Gw4Z · 2026-03-09

**Soundness:** 2
**Presentation:** 3
**Significance:** 3
**Originality:** 3
**Overall Recommendation:** 4
**Confidence:** 5

**Summary:**

This paper proposes a clustering-based approach to improve LLM-based watermarking (KGW) by addressing retokenization errors introduced during audio re-encoding. The method leverages community detection, particularly the Leiden algorithm, to cluster tokens that are frequently confused during retokenization, and then applies KGW-style watermarking at the cluster level. Experiments on the Moshi model show significantly improved detectability compared to the raw KGW and WMAR baselines, while maintaining comparable audio quality and demonstrating improved robustness under various non-temporal perturbations.

**Compliance With Llm Reviewing Policy:**

Affirmed.

**Final Justification:**

The rebuttal addressed my main concerns. I will keep my score since it's positive.

**Key Questions For Authors:**

1. How is the TPR@FPR under attack settings? I would like to see some empirical results. There are only log(p) values in Table 3 and Table 4.

2. Why does the proposed method perform much worse when prompted with LibriSpeech samples? How does the proposed method perform on other TTS models (e.g., encoder–decoder architectures such as FireRedTTS-1s, SparkTTS, etc.)?

3. The proposed method appears to be limited to encoder–decoder architectures. However, many recent TTS models use flow matching as a solution. How can the proposed method be transferred to flow-matching-based models such as the CosyVoice series?

**Limitations:**

Yes.

**Strengths And Weaknesses:**

Strengths:

1. The paper is well written and easy to follow.

2. The proposed clustering approach is simple yet effective. Intuitively, clustering tokens that are frequently confused during retokenization helps mitigate the impact of retokenization errors on watermark detection.

3. The experiments evaluate the method under a variety of attack settings, providing a comprehensive assessment of robustness.

Weaknesses:

1. The worse performance on LibriSpeech samples is not explained.

2. The proposed method is limited to encoder–decoder architectures. However, some recent methods use flow matching.

3. The evaluation is limited to two models. More state-of-the-art models should be considered.

---

> ### Author Rebuttal · Authors · 2026-03-31
>
> Thank you for your kind and insightful comments, acknowledging our method’s simplicity, and extensive robustness evaluation. We would like to fully address the identified weaknesses and clarify your questions.
>
> **W1**. LibriSpeech performance:
> We attribute the worse performance on LibriSpeech due to the Moshi model's training distribution and scope as a conversational model, while LibriSpeech prompts are audiobook excerpts.
>
> **W2** and **W3**. Extension to flow-matching methods and more models:
> Extension to many state-of-the-art models that use flow-matching for acoustic refining is definitely possible, as long as there is a discrete vocabulary (from an autoregressive semantic model). In this paradigm, flow matching can be considered part of the decoder shown in Figure 2. We experiment with CosyVoice3, as prompted by your **Q3**, as well as the most popular of the two proposed architectures in **Q2**, Spark-TTS:
> | Model (method)  |   Median p |   Mean -log(p) | FAD   |
> |:------|---:|-----:|:------|
> | Cosyvoice (Unwatermarked) |  0.2256    |    0.786 | 0.0137|
> | Cosyvoice (KGW)   |  0.2166    |    0.984 | 1.6525|
> | Cosyvoice (Ours)  |  3.334e-16 |   15.887 | 0.6156|
> | Spark-TTS (Unwatermarked) |  0.6095    |    0.358 | 0.2535|
> | Spark-TTS (KGW)  |  1.57e-07  |    7.108 | 0.2312|
> | Spark-TTS (Ours) |  8.464e-15 |   14.368 | 0.2675|
>
> Undoubtedly, our method is highly versatile and boosts the detectability of token-based watermarks by several orders of magnitude, without severely compromising quality. We note that while the existing setting of $\delta=2$ (used for Moshi) works fine for Spark-TTS, we had to attenuate $\delta$ down to 0.5 to not get much distortion with CosyVoice
>
> **Q1**. TPR@FPR under attacks:
> We apologize for not presenting TPR@FPR under attack settings, as per standard practice in the literature. We chose to focus on the average order of magnitude of the $p$-values as a better proxy for detectability. Below we clarify the TPR@FPR (for FPR=1% and FPR=0.1%) in Tables 3 and 4 on the existing experiments. First, for audio prompts (Table 3):
> | Attack    | Base 1% | Base .1% | WMAR 1% | WMAR .1% | WMAR (aug) 1% | WMAR (aug) .1% | Ours 1% | Ours .1% |
> |:----------|------:|----:|------:|-------:|--------:|-------:|------:|------:|
> | Identity  |  99.0 |  97.5 | 100.0 |   99.5 |    99.0 |   99.0 |  100.0 |  100.0 |
> | Smooth    |  45.0 |  17.0 |  28.0 |   10.5 |    85.0 |   57.0 |  100.0 |  100.0 |
> | Lowpass   |  98.5 |  93.0 |  99.5 |   99.5 |    99.5 |   99.0 |  100.0 |  100.0 |
> | Highpass  |   9.5 |   2.5 |  22.5 |    4.0 |    45.5 |   12.0 |  100.0 |  100.0 |
> | Noise     |  49.0 |  23.0 |   1.0 |    0.0 |    98.5 |   97.0 |   99.5 |   99.0 |
> | MP3       |  98.5 |  96.5 | 100.0 |   99.5 |    99.0 |   99.0 |  100.0 |  100.0 |
> | DAC       |  98.5 |  96.0 | 100.0 |   99.0 |    99.0 |   97.5 |  100.0 |  100.0 |
> | EnCodec   |  58.5 |  32.5 |  74.0 |   39.5 |    71.5 |   43.5 |  100.0 |  100.0 |
> | SpeechTok |  93.0 |  73.5 |  92.5 |   77.5 |    90.0 |   74.0 |  100.0 |  100.0 |
> | FaCodec   |  96.0 |  77.0 |  97.0 |   90.0 |    95.5 |   81.5 |  100.0 |  100.0 |
> | Crop      |  25.5 |   9.5 |  16.0 |    6.0 |    23.5 |   12.5 |   99.5 |   99.5 |
> | Shift     |  35.5 |  15.5 |  40.2 |   12.0 |    47.5 |   32.0 |  100.0 |  100.0 |
> | Speedup   |  24.0 |   6.5 |  16.0 |    4.0 |    20.0 |    5.5 |  100.0 |  100.0 |
>
> And for LibriSpeech (Table 4):
> | Attack.   | Base 1% | Base .1% | WMAR 1% | WMAR .1% | WMAR (aug) 1% | WMAR (aug) .1% | Ours 1% | Ours .1% |
> |:----------|------:|------:|------:|-------:|--------:|-----:|-----:|------:|
> | Identity  |  69.6 |  49.4 |  92.3 |   83.3 |    85.1 | 75.0 | 99.4 |  98.8 |
> | Smooth    |  26.2 |   6.5 |  13.1 |    2.4 |    39.3 | 19.0 |100.0 |  98.8 |
> | Lowpass   |  67.3 |  42.9 |  85.1 |   66.7 |    76.2 | 63.7 |100.0 |  99.4 |
> | Highpass  |   5.4 |   1.2 |  12.5 |    1.8 |    33.9 |  8.9 |100.0 |  99.4 |
> | Noise     |  23.2 |   9.5 |   0.6 |    0.0 |    70.2 | 54.2 | 88.1 |  79.8 |
> | MP3       |  68.5 |  45.2 |  91.1 |   82.7 |    83.3 | 73.2 | 99.4 |  99.4 |
> | DAC       |  63.7 |  44.0 |  79.2 |   64.3 |    84.5 | 66.1 | 99.4 |  98.8 |
> | EnCodec   |  32.1 |   9.5 |  35.1 |   12.5 |    35.7 | 14.3 |100.0 |  98.2 |
> | SpeechTok |  51.2 |  28.0 |  53.6 |   29.2 |    47.6 | 28.6 | 99.4 |  98.8 |
> | FaCodec   |  56.5 |  30.4 |  54.8 |   32.7 |    58.9 | 32.1 |100.0 |  98.8 |
> | Crop      |  15.5 |   3.6 |  14.3 |    3.0 |    20.8 |  6.5 | 92.9 |  85.7 |
> | Shift     |  19.9 |   6.0 |  24.1 |    9.5 |    33.9 | 17.6 | 99.4 |  97.3 |
> | Speedup   |   8.9 |   1.8 |   7.7 |    0.6 |    16.1 |  3.6 | 99.4 |  97.6 |
>
> **Q2**. LibriSpeech performance and other encoder-decoder TTS models:
> Regarding the LibriSpeech performance, we kindly point out our response to **W1**.
> For performance on other TTS models, we kindly point out our response in **W2** and **W3**.
>
> **Q3**.
> We kindly point out our response in **W2** and **W3**.

---

> > ### Author Rebuttal · Reviewer_Gw4Z · 2026-04-03
> >
> > Thank you for your response. I will keep my score since it's positive.

---

> > > ### Author Response · Authors · 2026-04-08
> > >
> > > We are glad your concerns were addressed. Thank you once again for your positive assessment and your support.

---

### Official Review · Reviewer_fke6 · 2026-03-13

**Soundness:** 2
**Presentation:** 2
**Significance:** 3
**Originality:** 3
**Overall Recommendation:** 4
**Confidence:** 2

**Summary:**

The paper introduces a novel, gradient-free token-level watermarking designed for AR audio generation. A major challenge in watermarking continuous audio is the retokenization mismatch caused by the instability of neural audio codecs, which degrades the watermark signal. This paper leverages the inherent vocabulary redundancy of the discretization process. The authors propose capturing retokenization errors to build a confusion matrix, and then applying community detection algorithm to group frequently confused tokens into semantic clusters. By applying the standard KGW  logit biasing scheme at the cluster level rather than the individual token level, the watermark becomes highly robust to discretization noise.

**Compliance With Llm Reviewing Policy:**

Affirmed.

**Final Justification:**

The author partially resolved my initial concern but did not address my new one. Although I have not lowered my Overall Recommendation score, I have developed some doubts regarding the soundness of the paper.

**Key Questions For Authors:**

1. How does the watermark's detectability degrade if the model is prompted to generate out-of-domain audio (e.g., speech with complex background noise), where the retokenization error distribution might shift dramatically?
2. Have you experimented with a sliding window hash that aggregates information across multiple channels to increase security?
3. Have you analyzed the computational overhead during the detection phase?

**Limitations:**

Is it applicable to more types of TTS and audio generation methods?

**Strengths And Weaknesses:**

Strengths:
1. The core premise of addressing physical codec instability through structural vocabulary distillation rather than parameter updates is highly sound and computationally efficient.
2. The experiments demonstrate that the cluster-level watermark achieves detectability several orders of magnitude higher than both standard KGW and finetuned baselines WMAR under extremely low FPR.

Weaknesses:
1. I'm quite concerned about the impact on audio quality, as the current metrics are not comprehensive enough—lacking speech WER, PESQ, STOI, and subjective evaluations.
2. It remains unclear whether mainstream TTS methods such as Seed TTS and CosyVoice, which employ an LLM + flow-matching approach followed by tokenization, can still be effectively utilized.

---

> ### Author Rebuttal · Authors · 2026-03-31
>
> Thank you for your positive feedback about our proposed method’s soundness and improved detectability. We would like to address the weaknesses you pointed out and respond to your questions.
>
> **W1**. Quality metrics:
> You are right to question the audio quality given the relative unreliability of MOS estimators. PESQ and STOI are intrusive methods and do not apply in our generative setting since there is no ground truth waveform. However, we can calculate WER/CER between Whisper-based transcriptions of Moshi's audio, and its own text generation as reference. First, for audio prompts:
> | h |Method|   WER |  CER |
> |--:|:-----|------:|-----:|
> | - | Clean| 0.220 | 0.164 |
> | 0 | Base | 0.231 | 0.163 |
> | 0 | WMAR (aug) | 0.236 | 0.169 |
> | 0 | WMAR | 0.247 | 0.180 |
> | 0 | Ours | 0.299 | 0.214 |
> | 1 | Base | 0.250 | 0.178 |
> | 1 | WMAR (aug) | 0.248 | 0.177 |
> | 1 | WMAR | 0.247 | 0.180 |
> | 1 | Ours | 0.244 | 0.174 |
> | 2 | Base | 0.261 | 0.187 |
> | 2 | WMAR (aug)| 0.253 | 0.180 |
> | 2 | WMAR | 0.267 | 0.195 |
> | 2 | Ours | 0.240 | 0.171 |
>
> And for LibriSpeech:
> | h |Method|   WER |   CER |
> |--:|:-----|------:|------:|
> | - | Clean| 0.374 | 0.303 |
> | 0 | Base | 0.363 | 0.279 |
> | 0 | WMAR (aug) | 0.366 | 0.284 |
> | 0 | WMAR | 0.339 | 0.266 |
> | 0 | Ours | 0.447 | 0.367 |
> | 1 | Base | 0.454 | 0.360 |
> | 1 | WMAR (aug)| 0.440 | 0.353 |
> | 1 | WMAR | 0.428 | 0.339 |
> | 1 | Ours | 0.412 | 0.339 |
> | 2 | Base | 0.414 | 0.324 |
> | 2 | WMAR (aug) | 0.402 | 0.317 |
> | 2 | WMAR | 0.408 | 0.322 |
> | 2 | Ours | 0.373 | 0.289 |
>
> There is little deviation from unwatermarked generation, on par with the other watermarked baselines. Indeed, there is no significant audible difference between unwatermarked and watermarked samples. We kindly point out that we have included such samples in the supplementary material.
>
> **W2**. Application to LLM followed by flow-matching models:
> Our method relies on the effective clustering of retokenization errors, which would also be available in such architectures. We experiment with CosyVoice3 (since SeedTTS is not publicly available), showing that our method successfully generalizes:
> | CosyVoice3 |  Median p | Mean -log(p) |
> |:-----------|----------:|-------------:|
> | KGW        |  0.2166   |        0.984 |
> | Ours       | 3.334e-16 |       15.887 |
> We kindly point out that this also addresses the **Limitation** included in the review.
>
> **Q1**. OOD generalizability:
> While evaluation for additive Gaussian noise in Tables 3, 4, and 6 of the paper partly covers the influence of background noise, we did not explicitly consider an in-generation distribution shift. Unfortunately Moshi does not accept style prompts, and our attempts to prompt MusicGen to generate noisy music were unsuccessful. However, the proposed CosyVoice model uses an audio style prompt, so we managed to generate muffled speech by using a static noise style prompt:
> | CosyVoice3 (noisy) |  Median p | Mean -log(p) |
> |:-------------------|----------:|-------------:|
> | KGW                |  0.1574   |       1.013  |
> | Ours               | 2.182e-16 |      16.298  |
>
> While the audio sounds muffled, the detectability is close to the clean results under **W2**. We attribute this to good acoustic tokenization of CosyVoice, even in noisy settings.
>
> **Q2**. Multichannel hash:
> As we point out in Appendix D, in a multichannel tokenizer like Mimi and EnCodec, we choose to keep the hash of the first channel's context as key for every channel, since the first channel is more stable in retokenization. Using a multichannel context would not be significantly different than just increasing the context size $h$. We ran an experiment with a multichannel hash, aggregating the last token of the first $h$ channels and comparing it with the baseline of using the last $h$ tokens of the first channel:
> | Moshi     |  Median p | Mean -log(p) |
> |:----------|----------:|-------------:|
> | Multichannel  h=0 | 3.551e-44 |       42.281 |
> | Multichannel h=1 | 1.482e-12 |       11.85  |
> | Multichannel h=2 | 1.298e-09 |        8.774 |
> | Base h=0  | 1.011e-43 |       40.612 |
> | Base h=1  | 3.52e-22  |       20.813 |
> | Base h=2  | 4.082e-10 |        9.472 |
>
> Indeed, the watermark strength is relatively similar, but weakens faster in the multichannel setting due to the higher channel's larger retokenization error. In practice, we do not believe that this would be more effective against spoofing attacks, but given a specific threat model we could potentially devise a more secure configuration (e.g. the higher RVQ codebooks not being publicly available).
>
> **Q3**. Detection overhead:
> Any token-based method requires a forward pass of the waveform through the speech tokenizer. This is computationally similar to processing a waveform with any trained post-hoc watermark detector. We present the minimal detection duration below:
> | Moshi | Mean -log(p) | Time (s) |
> |:------|-------------:|:---------|
> | KGW   |        7.737 | 0.130    |
> | Ours  |       41.769 | 0.127    |

---

> > ### Author Rebuttal · Reviewer_fke6 · 2026-04-02
> >
> > Thank you for your response. I still have some questions.
> >
> > Why are the calculated WER and CER so high? Is it due to the generally poor generation quality of Moshi itself or the insufficient capability of the Whisper-based model used?
> >
> > When h=0, the proposed method shows a significant increase in WER and CER in the LibriSpeech scenario—and this is under the condition of RVQ multi-codebooks. When the proposed method is applied to semantic tokens with higher information density, such as those in W2's CosyVoice, will it have an even greater impact?
> >
> > The response in W2 only mentions detectability but does not provide any data on its effect on WER/CER and speech generation quality (NISQA/DNSMOS).  Besides, I noticed that in the responses to other reviewers (for Cosyvoice & Spark-TTS), FAD was reported, but key metrics for speech quality were omitted. Why is that?
> >
> > My understanding is that CosyVoice's semantic tokens contain both semantic and stylistic information. When these tokens are altered, could the impact on intelligibility and style consistency be more pronounced compared to RVQ multi-codebooks like Moshi? If so, this would limit the general applicability of the proposed method.

---

> > > ### Author Response · Authors · 2026-04-08
> > >
> > > Thank you for the thorough evaluation of our response. We will clarify and address all your concerns below.
> > > 1. Indeed, the ASR metrics are higher than expected. Upon inspection of the Moshi's logs and Whisper transcriptions, we found that we were reading the reference transcription wrongly. We fixed the issue in our ASR code and repeated the experiments, producing reasonable error rates for all models. Regarding Whisper, we used the large-v3 model (following Seed-TTS-eval, used for evaluating both proposed TTS models). In the following table we repeat the detectability versus quality experiments for Moshi ($h$=0), using the conversational speech prompts:
> > > | Method    | Median p |Mean -log(p)|FAD (VGGish)|FAD (CLAP)| NISQA|DNSMOS|    WER |    CER |
> > > |:----------|---------:|-----------:|-----------:|---------:|-----:|-----:|-------:|-------:|
> > > | Clean     |0.2609    |       0.682|      0.2122|   0.0668 | 3.55 |  4.38| 0.1987 | 0.1561 |
> > > | Base      |3.86e-09  |       7.853|      0.3659|   0.0750 | 3.66 |  4.53| 0.2256 | 0.1667 |
> > > | WMAR (aug)|2.374e-14 |      13.392|      0.4440|   0.1229 | 3.64 |  4.51| 0.2311 | 0.1672 |
> > > | WMAR      |5.091e-17 |      16.336|      0.7041|   0.1077 | 3.36 |  4.51| 0.1250 | 0.0866 |
> > > | Ours      |2.127e-44 |      42.269|      0.2776|   0.0783 | 3.69 |  4.44| 0.2265 | 0.1616 |
> > >
> > >    And again, using LibriSpeech prompts:
> > >    | Method    | Median p |Mean -log(p)|FAD (VGGish)|FAD (CLAP)| NISQA|DNSMOS| WER   | CER   |
> > >    |:----------|---------:|-----------:|-----------:|---------:|-----:|-----:|------:|------:|
> > >    | Clean     |0.3076    |       0.667|    0.9076  |  0.2109  | 3.06 |  3.68|0.2359 | 0.1843|
> > >    | Base      |0.0003441 |       4.645|    1.9773  |  0.1954  | 3.02 |  4.02|0.3511 | 0.2894|
> > >    | WMAR (aug)|9.865e-06 |       6.689|    2.1770  |  0.2360  | 3.00 |  4.00|0.3339 | 0.2762|
> > >    | WMAR      |8.848e-09 |       9.504|    2.4836  |  0.2203  | 2.90 |  3.99|0.3150 | 0.2648|
> > >    | Ours      |8.797e-22 |      22.651|    1.2729  |  0.1887  | 3.19 |  3.84|0.2844 | 0.2198|
> > >
> > >    Notice that the detectability metrics and the non ASR quality metrics are very close to the original values in the paper (not exact match due to smaller subset for this run), since they had been calculated correctly.
> > >
> > > 2. The generative (not TTS) model Moshi appears to have relatively high speech recognition errors, but this holds for even the unwatermarked case. We inspect its outputs and attribute the higher error to slight misalignment of its own transcriptions with the audio. We attribute the higher WER/CER with LibriSpeech simply due to the model's poorer handling of non-conversational audio.
> > >
> > > 3. We do not notice significant degradation when our watermark is applied to semantic tokens, such as in the case of CosyVoice or Spark-TTS. Notice, many logit reweighting watermarks work on the purely semantic text domain. Furthermore, the frame size for tokenized speech is typically around tens of milliseconds (25 Hz for CosyVoice3), and can accommodate the stochasticity on the token sequence that stems from either a watermark or just top-k sampling.
> > >
> > > 4. Thank you for pointing out the confusing metric reporting, for which we apologize. We omitted the metric included in the other responses merely by accident, and we focused on FAD only because it was the most informative quality metric in the original paper (showing some tradeoff with watermarking). Please allow us to present the full metrics below:
> > > | Model (method)           | Median p |Mean -log(p)|FAD (VGGish)|FAD (CLAP)|NISQA|DNSMOS| WER  |  CER |
> > > |:-------------------------|---------:|-----------:|-----------:|---------:|----:|-----:|-----:|-----:|
> > > | Cosyvoice (Unwatermarked)|0.1885    |      0.863 |  0.0964    |  0.0235  |3.86 |2.73  |0.0323|0.0178|
> > > | Cosyvoice (KGW)          |0.03394   |      1.564 |  0.1954    |  0.0267  |3.75 |2.70  |0.0586|0.0366|
> > > | Cosyvoice (Ours)         |4.89e-14  |     13.927 |  0.1942    |  0.0294  |3.81 |2.74  |0.0519|0.0308|
> > > | Spark-TTS (Unwatermarked)|0.4953    |      0.4   |  0.2057    |  0.0401  |3.37 |2.94  |0.0097|0.0018|
> > > | Spark-TTS (KGW)          |2.061e-09 |      9.237 |  0.2221    |  0.0484  |3.30 |2.97  |0.0098|0.0012|
> > > | Spark-TTS (Ours)         |5.466e-18 |     17.806 |  0.3506    |  0.0472  |3.46 |2.96  |0.0099|0.0024|
> > >
> > >    Overall, our method shows dramatic detectability gains without any compromise in quality. Once again, thank you for pointing out the metric reporting inconsistency, this will considerably improve the quality of our final paper.

---

### Official Review · Reviewer_WiDb · 2026-03-14

**Soundness:** 3
**Presentation:** 3
**Significance:** 3
**Originality:** 3
**Overall Recommendation:** 4
**Confidence:** 3

**Summary:**

This paper studies the problem of robust watermarking for synthetic audio generated by autoregressive models. The main motivation is that token-level in-generation watermarking, while effective in text, becomes much less reliable in continuous modalities because watermark detection depends on retokenizing the generated waveform, and this retokenization is often inconsistent. The paper proposes a simple and elegant solution: instead of watermarking at the raw token level, it clusters tokens according to retokenization confusion patterns and applies the watermark at the cluster level. The method is gradient-free and does not require codec finetuning or white-box access, which makes it appealing in practice. The paper also provides a statistical analysis of detectability under retokenization mismatch and evaluates the method on Moshi/Mimi and MusicGen/EnCodec settings.

**Compliance With Llm Reviewing Policy:**

Affirmed.

**Final Justification:**

I appreciate the authors’ effort in providing a detailed rebuttal. The responses have addressed my concerns, and I will raise my score to 4.

**Key Questions For Authors:**

Please see the weaknesses

**Limitations:**

yes

**Strengths And Weaknesses:**

Strengths
1. The paper identifies a clear and important failure mode in audio watermarking: retokenization mismatch can destroy the statistical signal used by token-level detectors. This makes the problem both well-motivated and practically relevant.
2. The proposed method is simple and elegant. It builds a confusion graph from retokenization behavior, applies community detection, and replaces token identities with cluster identities during watermark generation and detection.
3. The method is appealing from a practical perspective because it is lightweight, gradient-free, black-box compatible, and easy to integrate into existing KGW-style watermarking frameworks.
4. The paper presents a satisfying connection between theory and experiment. The theoretical analysis argues that the improvement comes from replacing the token match rate with a larger cluster match rate, and the empirical results support this intuition.

Weaknesses
1. The theoretical analysis relies on simplified assumptions about retokenization noise, which may not fully capture temporal or cross-channel dependencies in real codecs. More empirical validation of these assumptions would strengthen the paper.
2. The tradeoff introduced by clustering is not fully characterized. Although the paper notes that coarser clustering can increase robustness at the cost of reduced key space, more collisions, and possible entropy loss, this tradeoff is not studied systematically enough.
3. The claim of cross-model generality would be more convincing if the main paper included more prominent results beyond Moshi, since much of the main experimental presentation is centered on that setting.
4. Temporal misalignment remains a notable limitation. While the proposed method is more robust than the baselines, cropping, shifting, and speed changes still substantially disrupt token-level watermark detection.

---

> ### Author Rebuttal · Authors · 2026-03-31
>
> Thank you for the detailed and thoughtful review, we appreciate your acknowledgement of our method’s elegance, practicality, and theoretical motivation. We are happy to fully address the weaknesses you pointed out.
>
> **W1** Validation of theoretical analysis:
> In order to strengthen the validity of our analysis, we empirically verify the expected $z$-score for single-channel watermarking (derived in Section 3.3). We use MusicGen, since we were able to embed robust watermarks on only one of its RVQ channels (unlike Moshi, where watermarking on only one of the 8 channels was not strong). First, we estimate $r$ and $g$ from an independent set of samples using Eq. (2) and (3). Then we proceed to estimate Eq. (5) for a fixed generation length $N$, comparing it with the actual empirical $z$-score from a set of samples (independent from the previous ones):
> | $\gamma$ | h=0 Theor. | h=0 Emp. | h=1 Theor. | h=1 Emp. | h=2 Theor. | h=2 Emp. |
> |---|----|---|-----|----|----|---|
> | 0.1   | 6.1123  | 6.7318   | 4.9797  | 9.2911   | 4.0570  | 7.6467 |
> | 0.2   | 7.0953  | 7.0225   | 5.7805  | 7.5169   | 4.7094  | 6.8360 |
> | 0.3   | 6.8406  | 6.8239   | 5.5730  | 7.5557   | 4.5404  | 6.0868 |
> | 0.4   | 6.3152  | 5.6013   | 5.1450  | 6.3814   | 4.1916  | 5.7638 |
> | 0.5   | 5.9394  | 5.3666   | 4.8389  | 5.6507   | 3.9422  | 4.6116 |
> | 0.6   | 5.0018  | 4.5129   | 4.0749  | 4.9890   | 3.3199  | 4.2382 |
> | 0.7   | 4.3240  | 4.1575   | 3.5228  | 3.9197   | 2.8700  | 2.9864 |
> | 0.8   | 3.4383  | 3.5239   | 2.8012  | 3.0123   | 2.2821  | 2.3622 |
>
> We also calculate the correlation of the theoretical and empirical estimations for each $h$:
> | h  | Correlation |
> |----|----|
> | 0  |  0.9465  |
> | 1  |  0.8535  |
> | 2  |  0.8872  |
>
> The high correlation indicates that our modeling, while simplified, captures very well the interplay of token-level watermarking with an unstable tokenizer. Thus, boosting detectability via artificially increasing $r$ via clustering is theoretically sound. Unfortunately, the clustering effect is not as easily predictable, and we are unable to verify Eq. (7) with equally high correlation. We attribute this to $g$ (the aggregate effect of watermarking) implicitly relying on the clustering map used, and thus being intractable to model. However, our derivation is still valid as a tractable simplified model indicating how watermark parameters are expected to impact the detectability.
>
> **W2** Full characterization of the clustered vocabulary tradeoff:
> Indeed, from LLM watermarking literature, a smaller vocabulary will inadvertently cause key collisions, which will result in repetitive green lists, leading to biased and degraded generation. In our paper we discuss and acknowledge this tradeoff, but do not fully characterize it, nor do we claim to achieve a Pareto-optimal solution, due to its intractability and complexity. We simply use retokenization errors as a practical heuristic for meaningful token clustering, and showcase that it works very well experimentally. We should note that the $K_{min}$ introduced in Eq. (8) is an arbitrary threshold that can be selected empirically, and the equation itself just showcases how the need for clustering (robustness) and context size (security) are competing variables.
>
> **W3** Cross-generality:
> Our results focused on two multichannel codecs with similar architecture. However, given the suggestions of the other reviewers, we successfully validated our method for the CosyVoice3 and Spark-TTS models. We used the same default parameters as for Moshi, and adjisted $\delta=0.5$ for CosyVoice:
> | Model (method)  |   Median p |   Mean -log(p) | FAD |
> |:----|----:|---:|:----|
> | Cosyvoice (Unwatermarked) |  0.2256    |    0.786 | 0.0137|
> | Cosyvoice (KGW)       |  0.2166    |    0.984 | 1.6525|
> | Cosyvoice (Ours)    |  3.334e-16 |   15.887 | 0.6156|
> | Spark-TTS (Unwatermarked) |  0.6095    |    0.358 | 0.2535|
> | Spark-TTS (KGW)     |  1.57e-07  |    7.108 | 0.2312|
> | Spark-TTS (Ours)   |  8.464e-15 |   14.368 | 0.2675|
>
> The results showcase that our method is highly general across models, and boosts the detectability of token-based watermarks by several orders of magnitude, with minimal impact on quality, as claimed in the paper.
>
> **W4** Temporal misalignment:
> You are right to point out this notable limitation. Indeed, any token-level watermark is inherently brittle to such attacks. However, synchronization methods such as the ones discussed in WMAR (Jovanović et al. 2025) and in our Appendix B are the only way to recover the original alignment and thus detect the watermark. For instance, running watermark detection on randomly shifted versions of the waveform can alleviate cropping and shifting attacks. Similarly, applying a grid search to estimate and correct for the speed change, can potentially realign the modified audio. We believe that in a practical application, an ensemble of post-hoc and token-based methods can be leveraged, since both have different strengths.

---

> > ### Author Rebuttal · Reviewer_WiDb · 2026-04-03
> >
> > I appreciate the authors’ effort in providing a detailed rebuttal. The responses have addressed my concerns, and I will raise my score to 4.

---

> > > ### Author Response · Authors · 2026-04-08
> > >
> > > We are truly glad that our responses addressed your concerns. Thank you so much for raising the score and your kind support for our work.

---

### Decision · Program_Chairs · 2026-04-30

**Decision:**

Accept (regular)

**Comment:**

The paper extends KGW watermarking for audio generation, where token-level watermarking is often unreliable due to inconsistent retokenization of continuous signals, offering a new gradient-free watermarking scheme. The reviewers generally support acceptance at the end of the rebuttal period, appreciating the simplicity, practicality, and well-supported empirical results.

The rebuttal was effective in addressing major concerns of the reviewers; while one reviewer noted some remaining concerns regarding soundness despite the rebuttal addressing earlier points, the overall consensus remains positive. AC agrees that the paper presents a simple yet effective idea with clear motivation and is likely to be of practical interest to the community.

AC recommends a weak acceptance, given the remaining room for improvement in the overall presentation and the reliability of the empirical evaluation. For instance, the authors identified and corrected an issue in their evaluation pipeline during the rebuttal phase, which somewhat reduces confidence in the reported results.

AC encourages the authors to clearly incorporate the corrected and additional experiments provided in the rebuttal into the revision.